# Simulation Study for the Adsorption of Carbon Disulfide on Hydroxyl Modified Activated Carbon

**DOI:** 10.3390/molecules28124627

**Published:** 2023-06-07

**Authors:** Xiangyu Cui, Penghui Li, Baohua Hu, Teng Yang, Haichao Fu, Shuai Chen, Xiaolai Zhang

**Affiliations:** 1School of Chemistry and Chemical Engineering, Shandong University, Jinan 250014, China; 202032358@sdu.edu.cn (X.C.); 202212239@sdu.edu.cn (P.L.); bhhu@valiant-cn.com (B.H.); 2Valiant Corporation Limited, 11 Wuzhishan Rd., YEDA, Yantai 264006, China; yangteng@valiant-cn.com (T.Y.); fuhaichao@valiant-cn.com (H.F.); chenshuai@valiant-cn.com (S.C.)

**Keywords:** carbon bisulfide, hydroxyl, activated carbon, adsorb, spread

## Abstract

In this study, grand canonical Monte Carlo simulations (GCMC) and molecular dynamics simulations (MD) were used to construct models of activated carbon with hydroxyl-modified hexachlorobenzene basic unit contents of 0%, 12.5%, 25%, 35% and 50%. The mechanism of adsorption of carbon disulfide (CS_2_) by hydroxyl-modified activated carbon was then studied. It is found that the introduction of hydroxyl functional groups will improve the adsorption capacity of activated carbon for carbon disulfide. As far as the simulation results are concerned, the activated carbon model containing 25% hydroxyl modified activated carbon basic units has the best adsorption performance for carbon disulfide molecules at 318 K and atmospheric pressure. At the same time, the changes in the porosity, accessible surface area of the solvent, ultimate diameter and maximum pore diameter of the activated carbon model also led to great differences in the diffusion coefficient of carbon disulfide molecules in different hydroxyl-modified activated carbons. However, the same adsorption heat and temperature had little effect on the adsorption of carbon disulfide molecules.

## 1. Introduction

The treatment of volatile organic compounds (VOCs) has been important for improving the ambient air quality in China in recent years [1,2,3]. Badische Anilin-und-Soda-Fabrik (BASF) claims that all organic chemicals with boiling points equal to or lower than 250 °C are VOCs. Carbon disulfide is not only an important component of VOCs, but is also a typical pollutant emitted by petrochemical enterprises. Carbon disulfide is not only highly toxic but also a potential precursor of PM 2.5 and ozone (O_3_) in ambient air. Therefore, preventing carbon disulfide from escaping into the atmosphere and polluting the environment is an urgent issue for manufacturers of insoluble sulfur. Because of the high toxicity of CS_2_, the maximum allowable concentration in the air is 0.005 g/m^3^; even a small amount of CS_2_ is not allowed to be discharged into the atmosphere. The symptoms of human poisoning are divided into acute mild poisoning, severe poisoning and chronic poisoning, depending on the amount of carbon disulfide entering the human body and the degree of harm caused. If skin even carelessly comes into contact with carbon disulfide, it can cause local erythema and even bullae [4].

Therefore, the removal of CS_2_ from the atmosphere, and especially suppression of its emission at the source of production, is a troublesome problem in industry. The authors E.F. Spink and Muller [5] suggested using sodium hydroxide aqueous solution as an absorbent to purify waste gas (hydrogen sulfide, carbon disulfide, etc.). They claim that the absorbed carbon disulfide is desorbed from the absorbent by heat, and is then condensed and returned to the production cycle. At the same time, the air outlet of the absorber is led to a tower filled with activated carbon for adsorbing and recovering the residual carbon disulfide. According to T. Morooka et al. [6] air containing carbon disulfide can be dissolved in alkaline aqueous solution under the influence of catalysts such as Pd/C, Pd/SiO_2_ or H_3_PO_4_/Al_2_O_3_, and is then used to treat the oxidative hydrolysis products (SO_2_, H_2_S and CO_2_) of carbon disulfide. This treatment will produce wastewater containing alkali metal sulfite, sulfide and carbonate. According to this method [6], sulfur-containing compounds are produced by burning cellulose in excess air at 800 °C, and some studies have suggested that sulfur-containing compounds (including CS_2_) should be catalytically oxidized at 300–500 °C [7,8]. In the above process, removing sulfur-containing compounds (including CS_2_) causes considerable energy consumption, thus wasting energy. Most scholars use chemical adsorption to remove CS_2_ from the air as in the above experiments. However, they produce many byproducts that are not easy to use, such as sulfur dioxide (SO_2_) and alkali metal sulfate, and the reaction temperature and energy consumption are high, so CS_2_ cannot be recycled during production, which further increases the production cost. Therefore, finding an efficient method to remove CS_2_ from the air has become an important issue for researchers and manufacturers.

Physical adsorption in porous materials is considered an energy-saving option in the absorption process of nanoporous carbon (DNC) because it does not require a large amount of solvent to be reboiled. DNC is synthesized from carbon-containing precursors, such as wood, coconut shell, coal, polymer and crystalline carbide [9,10,11,12]. Because of its micropore and mesopore characteristics and large specific surface area, DNC has shown excellent performance in gas and liquid phase separation applications. DNC is widely used in industrial production as adsorbents to reduce the emission of harmful substances, such as volatile organic compounds, dioxins and sulfates, into the external environment [10]. DNC can also be used as an active adsorbent, catalyst and catalyst carrier to degrade gaseous pollutants such as sulfide, NOx compounds and ammonia [13,14]. Filtration systems based on DNC also play an important role in removing biological pollutants, lead, mercury and arsenate from wastewater and drinking water [15,16,17].

The activated carbon adsorption process has the advantages of low operating cost, high mechanical strength and simple operation, but it also has some disadvantages. For example, to develop and optimize desulfurization and gas separation technologies, it is necessary to obtain balanced adsorption data from activated carbon materials; however, the isothermal adsorption of different components is difficult to determine. In addition, the functional groups on the surface of activated carbon also affect the adsorption process, which leads to blindness in the selection of activated carbon materials. In addition to experimental research, some researchers also use various simulation methods to predict and compensate for experimental defects. For example, Demir and Ahunbay [18] and others modeled the mesoporous structure of activated carbon by putting oxidized corals and coral fragments into a simulation box, and collecting independent graphite cracks. They then simulated the adsorption process of perfluorooctane sulfonate (PFOS) using the GCMC method, and it was found that the adsorption near the surface was stronger than that near the plane. Bahamon and Vega [19] et al. used GCMC to simulate the adsorption of five drug molecules in aqueous solution on activated carbon and found that the activated carbon model was highly selective to drugs, and the adsorption increased with increasing concentration in aqueous solution. Liu [20] et al. simulated the adsorption behavior of benzene on three kinds of activated carbons with different functional groups and found that the addition of functional groups provided many activation sites for benzene adsorption. Other studies show that the adsorption capacity of activated carbon under specific conditions can be changed by changing its functional groups and porous structure [21,22], and the modified activated carbon can be applied to industrial production to adsorb specific substances [23,24,25].

Although Monte Carlo method has some limitations, such as the deterministic problem needs to be transformed into stochastic problem, and the error is probability error, etc., molecular simulation is considered and the GCMC method in particular, can predict the adsorption properties of materials [26] and is a very suitable tool to supplement experimental work. By fixing the chemical potential, volume and temperature of the system, to calculate the ensemble average of a certain property of the giant canonical ensemble is shown in Formula (1) (Where *Z_μVT_* represents the holomorphic partition ratio and λ is the wave of de Broglie). The structural model of activated carbon is established in this paper, and the changes in adsorption behavior of CS_2_ on different structural models is introduced, including in response to change in pore volume and porosity, isosteric heat rejection, adsorption position, interaction between adsorbate and adsorbent, kinetics, diffusion coefficient, etc. The results can provide a useful reference for selecting a suitable CS_2_ adsorbent under specific conditions and developing new activated carbon adsorption materials. In addition, although it is found that the activated carbon model containing 25% hydroxyl modified activated carbon basic unit has the best adsorption performance and high adsorption efficiency at 318 K and atmospheric pressure, it can reduce energy consumption and production cost if applied to production. However, it has not been confirmed by experiments, and it may still be different from the reality.
(1)$$\langle A\rangle =\frac{\sum_{N=0}^{\infty }\frac{\lambda^{-3N}}{N!}exp\left(\beta N\mu \right)\int exp\left[-\beta E\left(r^{N}\right)\right]dr^{N}}{Z_{\mu VT}}$$

## 2. Model Construction and Simulation Details

### 2.1. Model Construction

The research of Juhola [27], Dollimore, Leonard [28,29] and others [30,31,32] has shown that the basic unit of activated carbon is the graphene monolith,, which are in turn composed mainly of carbon atoms arranged in a hexagonal aromatic ring structure with other carbon atoms at a distance of 0.142 nm [33]. It is generally believed that they may also contain pentagonal and other ring structures [34,35], which leads to the distortion of nanocrystals. At the same time, activated carbon also contains some heteroatoms, such as O, N, S and H. Wide-angle neutral density (ND) and transmission electron microscope (TEM) images show that AC generated at low temperature (<800 °C) is highly disordered, and the bond coordination numbers range from 2.0 to 2.5, indicating that the nanocrystals are limited in scope and highly defective [35], as shown in Figure 1c,d. The irregular stacking of nanocrystals in AC leads to disordered domains and various porous features, as shown in Figure 2. Pore size distribution (PSD) shows that disordered carbon may contain pores ranging in size from less than one nanometer to several hundred nanometers, and usually more than half of the total pore volume is located in micropores smaller than two nanometers [36].

Activated carbon is mainly composed of polycyclic aromatic hydrocarbons, non-six-membered rings and a variety of functional groups, and its internal structure is complicated. For this reason, a huge amount of calculation is needed to build a real activated carbon model, so some simplification is necessary. Therefore, in this study, four kinds of graphene monoliths (Figure 1) were constructed using the software Materials Studio 2018 (MS 2018) as the basic units of the activated carbon model and builds it according to literature [37]. The modification was realized by replacing the hydrogen atom at the edge of hexachlorobenzene (Figure 1a) with a hydroxyl group, and the modified hexachlorobenzene was marked as AC-OH (Figure 1b). Figure 1c,d show two kinds of graphene monoliths used to simulate the internal defects of activated carbon, with five-atom and seven-atom carbon rings, respectively. These four basic structural units are named AC, AC-OH, AC-5 and AC-7. Then, these four basic units are put in a box according to the methods of Muller group [38] and Sarkisov group [39], and hydroxyl modified activated carbon models with different porosity and pore size distribution can be constructed.

Then, using the Construction function of the Amorphous Cell module in the Materials Studio software, 80 of the above four basic units were filled into the cubic cell with three-dimensional periodic boundary conditions, and the target density was maintained at 0.55 g/cm^3^ [40]. At the same time, considering the defects of the carbon structure and to make the model more realistic, 5% graphene monoliths with five-atom rings and seven-atom rings were introduced into a single cell [37], that is, two basic units with two defects were included. Five kinds of activated carbon models with hydroxyl contents of 0%, 12.5%, 25%, 35% and 50% were constructed and named AC1, AC2, AC3, AC4 and AC5, respectively, as shown in Figure 2. The solvent-accessible surface area (S, Å^2^), porosity (P, %), limit diameter of pores (D_L_, Å) and maximum aperture (D_M_, Å) of the activated carbon models were determined. The values of these parameters are listed in Table 1.

### 2.2. Simulation Details

The adsorption process of CS_2_ on activated carbon was simulated using GCMC and MD and the activated carbon model [37]. First, energy minimization was adopted for its configuration, and the Forcite module was used. The settings were as follows: Task was set to energy; for Forcefield, the COMPASS force field commonly used in organic molecules was used; for Charges, we selected the Forcefield assigned; Electrostatic was set to the Ewald and Group addition method; and van del Waals was set to the atom-based addition method. Then, the same parameters were used to simulate the molecular dynamics of the five models. In this case, Task was set to dynamics, NVT ensemble was chosen for Ensemble to simulate the dynamics, Temperature was set at 318 K, step size was set to 1 fs, the total number of simulation steps was 10,000 and the total simulation time was 10 ps. After the calculation, a Monte Carlo simulation of adsorption behavior was carried out.

The radial distribution function (RDF) of the AC1 model can be obtained by using the Forcite Analysis module. As seen in Figure 3, the first peak of the AC1 model appears at approximately 0.13 nm, while the first peak of the AC1 model appears near 0.1 nm after adding functional groups, which shows that the introduction of functional groups reduces the pore structure of activated carbon. The peak value of the RDF curve is very consistent with most previous experimental data [40], and shows the rationality of the model and its similarity to the real activated carbon structure.

## 3. Results and Discussion

In this work, the grand canonical Monte Carlo simulation (GCMC) method was used to study the adsorption behavior and performance of activated carbon models with different hydroxyl contents for carbon disulfide. The GCMC program includes the transformation, rotation, destruction and generation of particles to ensure the potential chemical balance between the bulk phase and pore phase and the three-dimensional periodic boundary conditions (x, y and z). We used the Materials Studio 2018 (MS 2018) software and the Sorption module to calculate the adsorption performance. Task was set to Adsorption isotherm, and Method used the Metropolis algorithm to set the equilibrium adsorption configuration for the structural model system that adsorbs carbon disulfide molecules. The minimum and maximum fugacities of the system were set to 50 kPa and 200 kPa, respectively, to simulate the actual industrial production environment. The number of fugacity steps was set to 29, and the Temperature was maintained at 318 K. Forcefield used the COMPASS force field commonly used in organic molecules. Forcefield assigned was chosen for Charges, Electrostatic was set to the Ewald and Group addition method, and van del Waals used the Atom-based addition method. In the Properties tab of the Sorption Calculation dialog box, the Sample interval was set to 50 and the Grid interval to 0.4 Å.

After calculating the adsorption isotherm of the activated carbon model, the equilibrium adsorption configuration was calculated in the Sorption module. As long as Task is set to Fixed pressure and Fugacity is set to 101.3, we can obtain the equilibrium adsorption configuration adsorption configuration diagram of each activated carbon model at one atmospheric pressure, as shown in Figure 4. From Figure 4a, we can see the adsorption position and quantity of carbon disulfide molecules in different activated carbon models. The activated carbon units with hydroxyl groups adsorb more carbon disulfide molecules, which shows that the introduction of hydroxyl functional groups is beneficial to the adsorption of carbon disulfide molecules by activated carbon. It can also be seen from Figure 4b that the number of carbon disulfide molecules (in blue) is increased by the activated carbon basic unit (in red) in the AC1 to AC5 models.

### 3.1. Effect of Activated Carbon Models with Different Hydroxyl Contents on the Adsorption Isotherm of Carbon Disulfide at Atmospheric Pressure

Figure 5a shows the adsorption isotherm of carbon disulfide on activated carbon models with different hydroxyl contents at 318 K. As shown in Figure 5a, regardless of the kind of activated carbon model, the adsorption isotherm trend is the same, and can be roughly divided into three stages. In the first stage, the fugacity ranges from 50 kPa to 60 kPa, and the curve increases linearly with the highest slope. This means that the adsorption rate of carbon disulfide is also fast. In the second stage, the fugacity ranges from approximately 60 kPa to 100 kPa, and the slope of the adsorption curve drops sharply, indicating that the adsorption rate slows down. In the third stage, the fugacity ranges from 100 kPa to the simulated end point of 200 kPa. At this stage, the adsorption curve is almost parallel to the X axis, and the curve tends to stabilize gradually, which means that the adsorption of carbon disulfide is gradually balanced. The adsorption capacity of activated carbon to carbon disulfide increased with increasing hydroxyl content throughout the simulation process. As shown in Figure 5b, the AC5 model with 50% hydroxyl content adsorbed 579 carbon disulfide molecules, the most, of any model. This result shows that, in a certain range, the adsorption capacity for carbon disulfide becomes stronger with increasing hydroxyl content. As shown in Table 2, when the hydroxyl content is increased from 0% to 12.5%, an average of 2.88 carbon disulfide molecules can be adsorbed for every 1% increase in hydroxyl content. When the hydroxyl content increases from 12.5% to 25%, every 1% increase in hydroxyl content can absorb an average of 4.08 more carbon disulfide molecules. However, when the hydroxyl content is increased from 25% to 50%, only 3.6 carbon disulfide molecules can be adsorbed on average for every 1% increase in hydroxyl content. It can be seen from Table 2 that, when the hydroxyl content ranges from 0% to 25%, adsorption capacity increases and efficiency decreases with the increase in hydroxyl content. From the above analysis, it can be concluded that an increase in hydroxyl content enhances the adsorption capacity of activated carbon for carbon disulfide; however, when the hydroxyl content exceeds a certain amount, although the adsorption capacity of activated carbon for carbon disulfide continues to increase, the average amount of carbon disulfide adsorbed per 1% increase in hydroxyl content decreases, and the efficiency also decreases.

This may be because there is little interaction between activated carbon and carbon disulfide molecules under low pressure, and there are not enough carbon disulfide molecules to fill all the pores throughout the whole activated carbon model. At this time, porosity and pore size become important factors that affect the adsorption capacity of activated carbon. The ultimate diameter and maximum pore diameter of the internal pores of hydroxyl-containing activated carbon models AC2, AC3, AC4 and AC5 are obviously increased compared to those in the AC1 model without hydroxyl functional groups. The pores between the basic units of activated carbon are smaller, which makes the activated carbon structure more compact (see Table 1). This shows that the introduction of oxygen provides more adsorption sites on the internal structure of activated carbon, and carbon disulfide molecules are polar molecules, which easily form hydrogen bonds [30]. Therefore, the greater the density of oxygen atoms, the higher the adsorption capacity of carbon disulfide molecules on activated carbon. The literature [41] found that the adsorption capacity of single iron atom to carbon disulfide is higher than that of iron cluster, which is similar to the view that the activated carbon model with 12.5% hydroxyl modified basic unit found in this study has the highest adsorption efficiency to carbon disulfide. In addition, another literature [42] found that there are different adsorption sites on the surface of phosphorus olefin black by density functional theory, which is also consistent with the findings of this study. In summary, the introduction of hydroxyl functional groups can increase the number of adsorption sites, thus improving the adsorption capacity of activated carbon for carbon disulfide.

### 3.2. Effect of Activated Carbon Models with Different Hydroxyl Contents on the Adsorption Isotherm of Carbon Disulfide at Low Pressure

During industrial production of insoluble sulfur (near 1 atm), the introduction of hydroxyl functional groups can improve the activated carbon model’s adsorption capacity for carbon disulfide, and its adsorption efficiency for carbon disulfide molecules first increases and then decreases. The adsorption isotherm of carbon disulfide by activated carbon models with different hydroxyl contents under low pressure (0~1 kPa) was verified, as shown in Figure 6. Overall, the adsorption isotherms of the AC1 and AC3 models are low from beginning to end, being 4 and 8 until the end of 1 kPa adsorption, respectively. The trends in the adsorption isotherms of AC2, AC4 and AC5 are almost the same; they all rise at a relatively fast rate while fluctuating. Below 0.1 kPa, the fluctuation rate is relatively fast, and the increase rate is relatively slow. At 0.1 kPa until the end of adsorption, the fluctuation rate becomes relatively slow, but the increase rate is relatively large. Until the end of adsorption, their adsorption capacities all remain at approximately 30. By comparing Figure 5 and Figure 6, it can be seen that the adsorption capacity is lower at 318 K, regardless of the type of activated carbon model. From the perspective of production and application, atmospheric adsorption can adsorb more carbon disulfide molecules than low-pressure adsorption.

### 3.3. Effect of Different Hydroxyl Content on the Equivalent Adsorption Heat of Carbon Disulfide by Activated Carbon Models

Equivalent adsorption heat is another key parameter for measuring the adsorption performance of activated carbon, and represents the thermal effect in the adsorption process. The equivalent change in adsorption heat for carbon disulfide in different activated carbon molecular models is shown in Figure 7. During the whole adsorption process, the trend in the overall adsorption heat with the increase in the sulfur content of hydroxyl functional groups is the same for each of the five models AC1 to AC5. The adsorption heat curve is the steepest when the pressure is less than 80 kPa, whereas it increases slowly when pressure is in the range of 80 to 200 kPa. Therefore, the overall trend in adsorption heat remains the same when hydroxyl functional groups are introduced alone. As was reported in [43], it is possible that different functional groups are introduced into the activated carbon model. At low pressure (0~1 kPa), the average adsorption heat of benzene molecules by activated carbon is highest for AC-COOH, followed by AC-O, AC-OH and AC-H, in descending order. It can be seen from the above research that it is of little significance to explore the influence of activated carbon models with different hydroxyl contents on the equivalent adsorption heat of carbon disulfide if a functional group is introduced separately.

### 3.4. Diffusion Coefficient of Carbon Disulfide in Activated Carbon Models with Different Hydroxyl Contents

The diffusion coefficient reflects the dynamic characteristics of permeated gas and polymer materials. The diffusion coefficient formula is shown in Formula (2), where ***d*** is the dimension of the system and ***N*** is the number of target molecules in the system. The obtained MSD data are plotted against time ***t***. As shown in Figure 8, with increasing hydroxyl content, the diffusion coefficient ***D*** = a/6 can be obtained by fitting this straight line.
(2)$$D=\frac{1}{2dN}\underset{t\to \infty \infty }{\lim}\frac{d}{dt}\sum_{i=1}^{N}\langle [r_{i}{\left(t\right)}^{2}-r_{i}{\left(0\right)}^{2}]\rangle$$

At the same time, the specific diffusion coefficient can be obtained after fitting. From the combination of Table 3 and Figure 8, we can see that the diffusion coefficient of carbon disulfide molecules in activated carbon first decreases, then increases, then decreases and then increases with the increase in hydroxyl content in the activated carbon model. The diffusion coefficient of carbon disulfide is large in AC3 because the introduction of hydroxyl groups leads to an increase in the porosity of activated carbon, and the limit diameter D_L_ and maximum pore diameter D_M_ of pores are large, which makes it easier for carbon disulfide molecules to be transported and diffused through the system. When the hydroxyl groups content is 35% (AC4), the diffusion coefficient of carbon disulfide in activated carbon is lower than that of AC3. This may be because, after the introduction of hydroxyl groups, the four models of AC2, AC3, AC4 and AC5 have the lowest pore diameter D_L_ and the largest pore diameter D_M_, and the content of hydroxyl groups with strong electronegativity increases. This limits the surface charges of activated carbon and enhances its interaction with carbon disulfide molecules. This shows that the factors affecting the adsorption of carbon disulfide by activated carbon not only include pore structure, but also the non-bond interactions between the molecules and the system. The diffusion coefficient is increased in AC5 (the model with the highest hydroxyl content) because the high porosity of 69% renders the molecular folding degree in activated carbon insufficient, and the ability to limit the diffusion movement of carbon disulfide molecules is weakened.

### 3.5. Effect of the AC3 Activated Carbon Model on Adsorption of Carbon Disulfide at Different Temperatures

To explore the influence of temperature on the adsorption of carbon disulfide by hydroxyl-modified activated carbon, we measured the adsorption isotherms and mean square shifts of carbon disulfide molecules in the AC3 activated carbon model at different temperatures (298 K, 308 K, 318 K and 328 K), as shown in Figure 9. It can be seen from Figure 9a that the trend in the adsorption isotherm remains almost the same at different temperatures, and the adsorption capacity is the largest at the adsorption end point (298 K). Figure 9b shows that there is little difference between the diffusion coefficients at 298 K, 308 K and 318 K. It can be inferred that temperature has little effect on the adsorption of carbon disulfide in the hydroxyl-modified activated carbon model.

### 3.6. Trajectory of a Single Carbon Disulfide Molecule in Activated Carbon Models with Different Hydroxyl Contents

To more intuitively observe the diffusion behavior of carbon disulfide in different activated carbon models, the coordinates of the geometric center of the toluene molecule were extracted from the atomic coordinate file generated during the simulation of CS_2_ adsorption, and a three-dimensional trajectory diagram of CS_2_ diffusion near the boundary was drawn within 0~500 ps. Twenty-one data points were taken every 25 ps, in which the red sphere represents the instantaneous position of the CS_2_ molecule and the red line segment represents the movement trajectory of the CS_2_ molecule, as shown in Figure 10. The diffusion behavior of CS_2_ molecules in the AC1, AC2, AC3 and AC5 models is similar; after a long period of vibration near a certain point, the CS_2_ molecule jumps to another point to continue the vibration for another long period. Macroscopically, CS_2_ molecules jump from one pore in the activated carbon to another through vibration. In the AC4 model, the molecular diffusion of CS_2_ appears as a long period of “vibration”. CS_2_ may also be hindered, causing the diffusion path to shift, which makes the entire carbon disulfide molecule undergo a small transition while vibrating, thus prolonging the transition time from one point to another. This is because the interaction between the adsorbate and adsorbent becomes increasingly obvious with increasing hydroxyl ratio. The above phenomenon also conforms with the diffusion behavior mechanism of gas molecules in polymer materials [44].

## 4. Conclusions

In this work, grand canonical Monte Carlo simulation (GCMC) and molecular dynamics simulation (MD) were combined to construct activated carbon models with hydroxyl-modified hexachlorobenzene basic unit contents of 0%, 12.5%, 25%, 35% and 50%, and the mechanism of adsorption of carbon disulfide by hydroxyl-modified activated carbon was studied. The following conclusions were drawn:(1)Using Materials Studio 2018 (MS 2018) software, an activated carbon model that is similar to reality can be constructed, and all the models and simulation methods in this work can be used to provide ideas for exploring issues related to activated carbon in the future.(2)Hydroxyl-modified activated carbon enhances the adsorption capacity of carbon disulfide molecules, and the hydroxyl content has a great influence on the adsorption of carbon disulfide. With the increase in the basic unit content of hydroxyl-modified hexachlorobenzene, the adsorption capacity of activated carbon also increases, whereas the adsorption efficiency first increases and then decreases. The adsorption capacity for second-rate carbon sulfide molecules is the largest when the basic unit content of hydroxyl-modified hexabenzobenzene is 50%, and when it is 25%, the adsorption efficiency is the highest.(3)The adsorption sites of carbon disulfide molecules in the activated carbon model change after hydroxyl functional groups are introduced. Carbon disulfide molecules are polar molecules, which easily form hydrogen bonds during the adsorption process. The atoms in the activated carbon have a stronger superposition effect on the charge of carbon disulfide, which is beneficial to adsorption in a low-pressure environment.(4)With the increase in the basic unit content of hydroxyl-modified hexachlorobenzene, the porosity and solvent-accessible surface area of the activated carbon models increased, while the ultimate diameter and maximum pore diameter of the pores changed to different degrees. The change in the internal pore structure of these activated carbon models also led to a great difference in the diffusion coefficients of carbon disulfide molecules in different hydroxyl-modified activated carbons.(5)Equivalent adsorption heat and temperature were found to have little effect on the adsorption of carbon disulfide in the hydroxyl-modified activated carbon model.(6)According to this simulation, at 318 K and atmospheric pressure, the activated carbon model containing 25% hydroxyl-modified activated carbon basic unit has the best adsorption performance and high adsorption efficiency for carbon disulfide molecules, which can reduce energy consumption and production cost when applied to production.

## Figures and Tables

**Figure 1 molecules-28-04627-f001:**
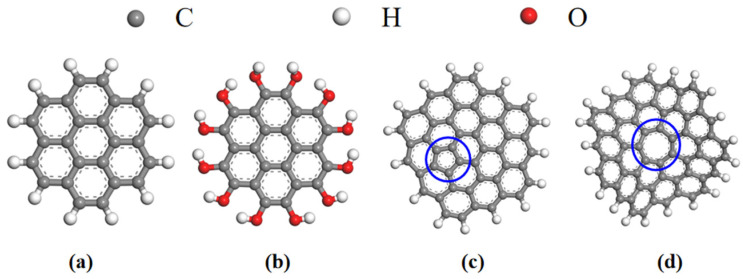
Basic unit of activated carbon used in this study: (**a**) hexabenzobenzene; (**b**) hydroxyl-modified hexabenzobenzene; (**c**) graphene monolith with a five-atom ring inside (The blue circle represents the defective monomer five-membered ring); (**d**) graphene monolith with a seven-atom ring inside (The blue circle represents the defective monomer seven-membered ring).

**Figure 2 molecules-28-04627-f002:**
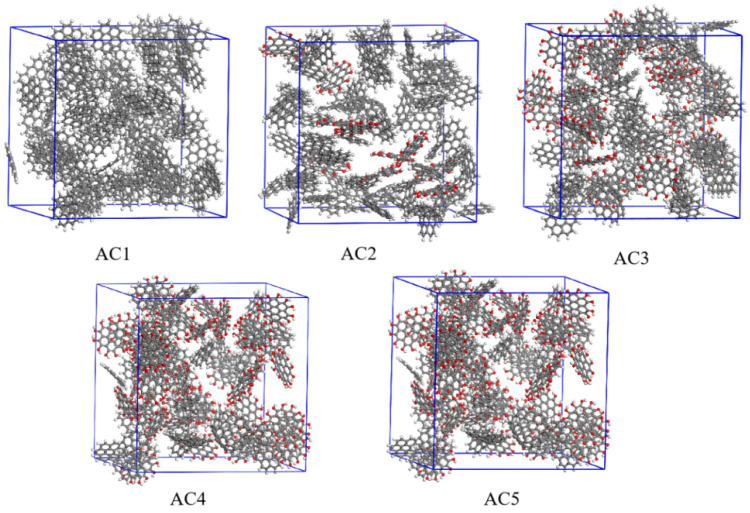
Basic configurations of five activated carbon models.

**Figure 3 molecules-28-04627-f003:**
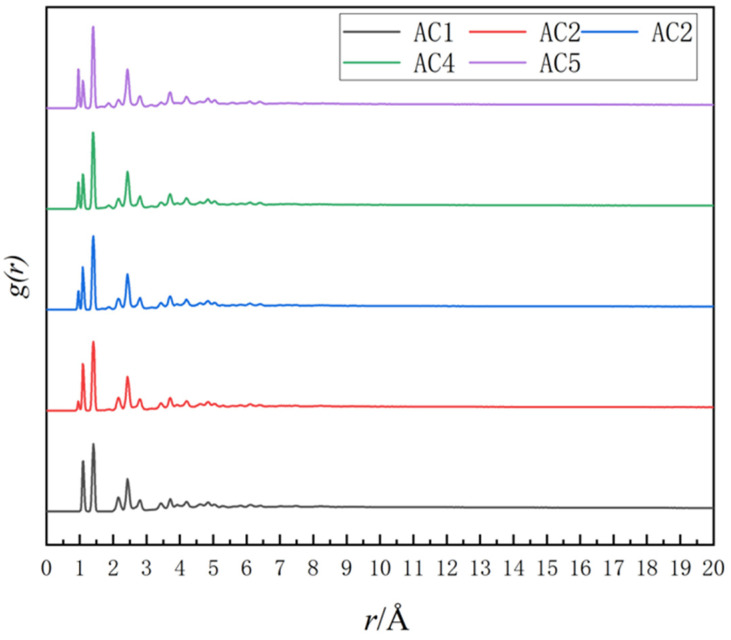
Radial distribution function of different activated carbon models.

**Figure 4 molecules-28-04627-f004:**
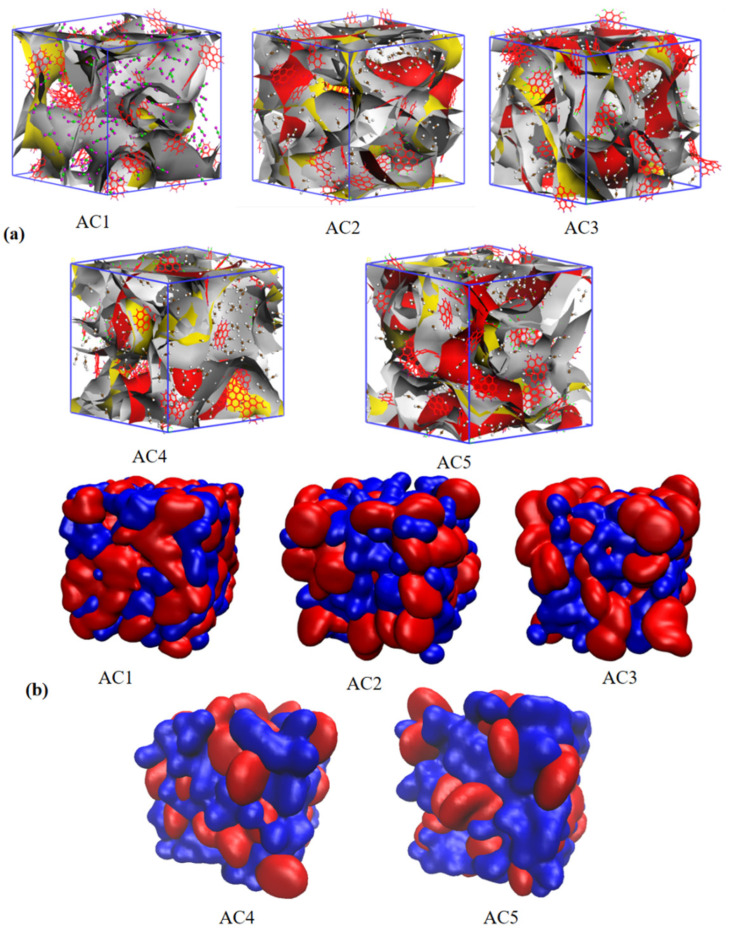
(**a**) shows the equilibrium adsorption configurations of different activated carbon models; (**b**) is a diagram of carbon disulfide in activated carbon with different hydroxyl contents (the basic unit of activated carbon is represented in red, and the carbon disulfide is represented in blue).

**Figure 5 molecules-28-04627-f005:**
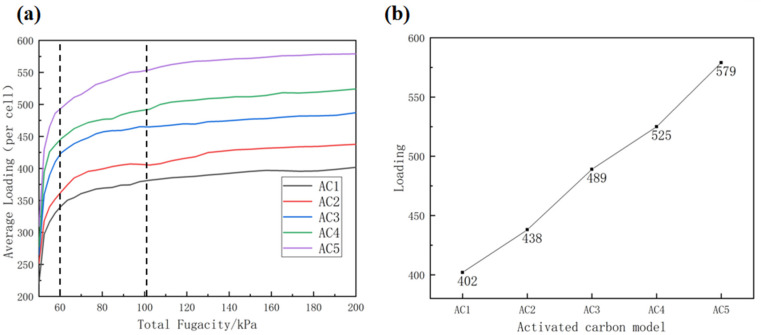
(**a**) Adsorption isotherms of carbon disulfide in activated carbon with different hydroxyl contents at atmospheric pressure; (**b**) Maximum loading of carbon disulfide in activated carbon with different hydroxyl contents.

**Figure 6 molecules-28-04627-f006:**
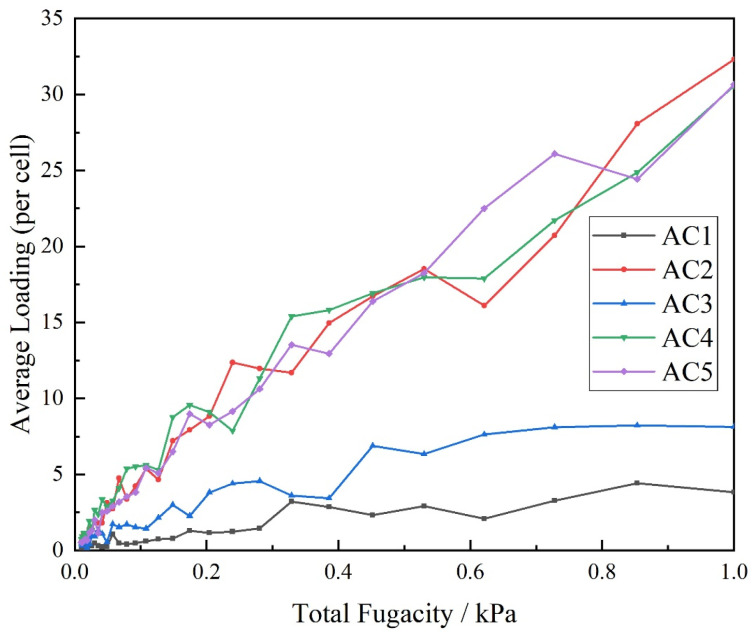
Adsorption isotherms of carbon disulfide in activated carbon with different hydroxyl contents at low pressure.

**Figure 7 molecules-28-04627-f007:**
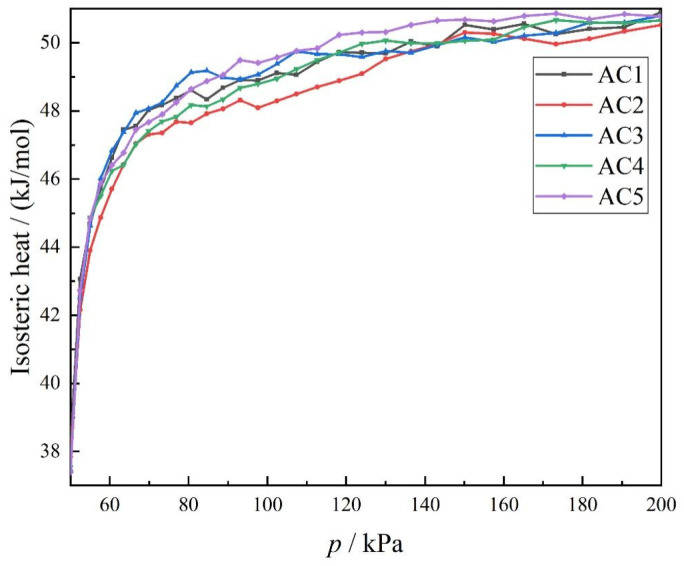
Equivalent adsorption heat of carbon disulfide in activated carbon models with different hydroxyl contents.

**Figure 8 molecules-28-04627-f008:**
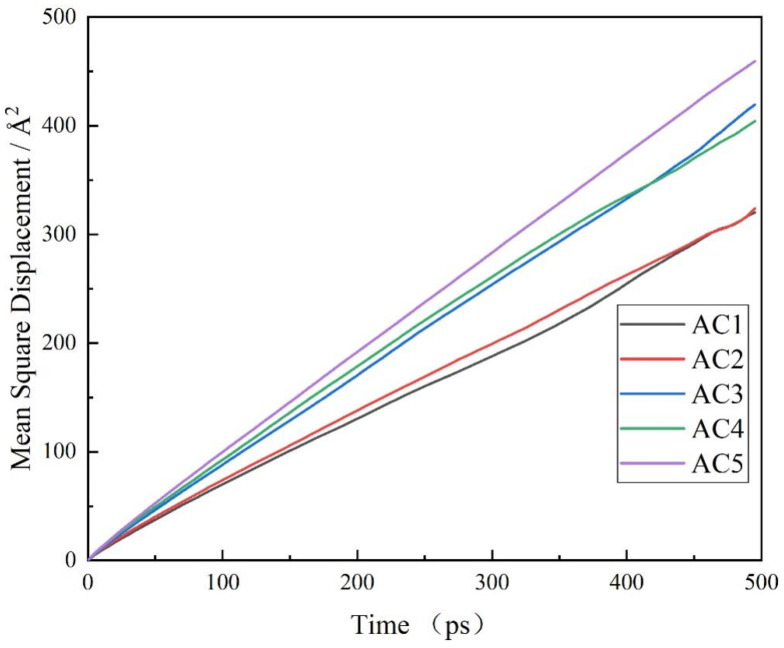
Mean square displacement of carbon disulfide in activated carbon with different hydroxyl contents.

**Figure 9 molecules-28-04627-f009:**
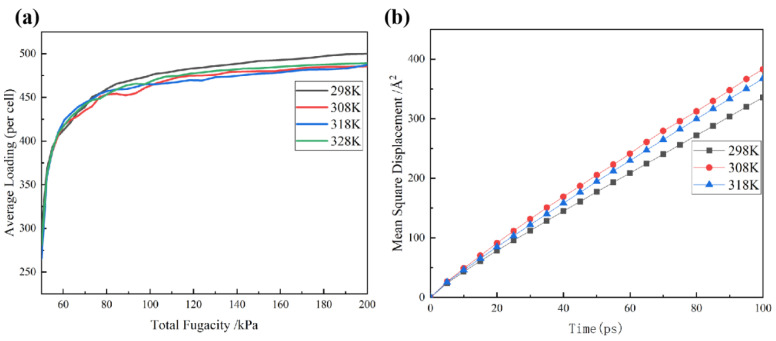
(**a**) Adsorption isotherms of carbon disulfide in the AC3 activated carbon model at different temperatures; (**b**) Mean square displacement of carbon disulfide in the AC3 activated carbon model at different temperatures.

**Figure 10 molecules-28-04627-f010:**
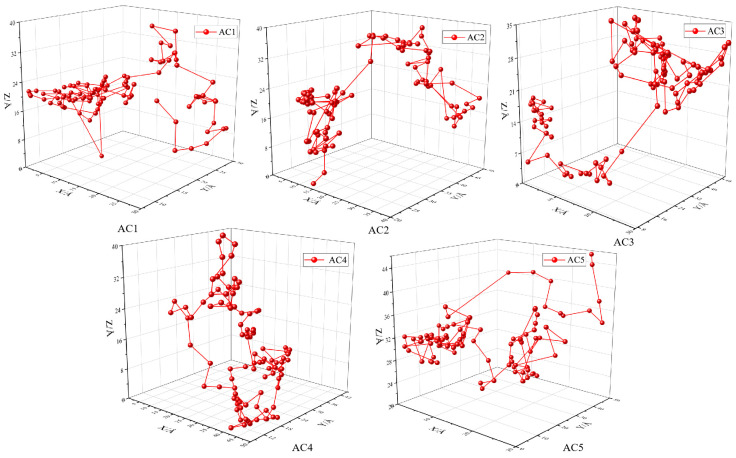
Movement trajectory of a single carbon disulfide molecule in activated carbon models with different hydroxyl contents.

**Table 1 molecules-28-04627-t001:** Detailed parameter settings of five activated carbon models.

Model	AC1	AC2	AC3	AC4	AC5
Contains basic units/number	AC: 76 AC-5: 2 AC-7: 2	AC: 66 AC-OH: 10 AC-5: 2 AC-7: 2	AC: 56 AC-OH: 20 AC-5: 2 AC-7: 2	AC: 48 AC-OH: 28 AC-5: 2 AC-7: 2	AC: 36 AC-OH: 40 AC-5: 2 AC-7: 2
Temperature/K	315	315	315	315	315
Box size/nm × nm × nm	4.26 × 4.26 × 4.26	4.36 × 4.36 × 4.36	4.46 × 4.46 × 4.46	4.54 × 4.54 × 4.54	4.65 × 4.65 × 4.65
Porosity P/%	69.93	71.11	73.62	73.48	75.53
Solvent can reach the surface area S/Å^2^	12,745.85	13,718.14	15,811.51	16,129.28	18,414.77
The limiting diameter of the hole D_L_/Å	6.15	8.21	8.03	7.44	7.78
Maximum aperture D_M_/Å	11.85	16.06	15.14	14.34	14.87

**Table 2 molecules-28-04627-t002:** Comparison of the adsorption capacity of carbon disulfide by different activated carbon models.

Hydroxy-modified hexaphenol content/%	0	12.5	25	35	50
Adsorption capacity at simulated end point/number	402	438	489	525	579
The average amount of carbon disulfide adsorbed by 1% hydroxyl/number	---	(438 − 402)/12.5 = 2.88	(489 − 438)/12.5 = 4.08	(525 − 489)/10 = 3.6	(579 − 525)/15 = 3.6

**Table 3 molecules-28-04627-t003:** Diffusion coefficient of carbon disulfide in activated carbon with different hydroxyl contents.

Model	AC1	AC2	AC3	AC4	AC5
Diffusion coefficient (×10^−5^cm^2^/s)	1.125	1.037	1.368	1.224	1.487

## Data Availability

The data are contained within the article.

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
