# Peer review of "Simulation Study for the Adsorption of Carbon Disulfide on Hydroxyl Modified Activated Carbon"

_molecules, 2023, doi:10.3390/molecules28124627_

Round 1

Reviewer 1 Report

Review report on the manuscript titled:

Study on the adsorption of carbon disulfide by hydroxyl-modified activated carbon

ID du manuscrit : molecules-2414975

This is an important paper on carbon disulfide adsorption by hydroxyl-modified activated carbon using simulation. This study cannot be published in this journal in its current state. I recommend the review below

- The title does not exactly reflect the work in the manuscript, I suggest: "Simulation study for the adsorption of carbon disulfide on hydroxyl modified activated carbon".

- The abstract and the conclusion need to be improved: it is necessary to add more quantitative results and a pseudo final conclusion.

- English needs serious revision by a native or professional.

- Try to avoid the use of acronyms at the beginning of the introduction without defining them for the first time.

- The introduction should be improved to show the purpose and gaps on which you are basing your study.

- The references are not complete throughout the manuscript, I suggest these studies to improve the part of the introduction devoted to activated carbons: https://doi.org/10.1016/j.jclepro.2023.136333,

https://doi.org/10.1016/j.biortech.2022.128162,

https://doi.org/10.1016/j.diamond.2023.109834

- It is necessary to check the units in the tables and figures

English needs serious  revision by a native or professional 

Reviewer 2 Report

Reviewer Report

In this manuscript, the authors report on the use of the grand canonical Monte Carlo simulation (GCMC) and molecular dynamics simulation (MD) to construct models of activated carbon with hydroxyl-modified hexa- 9 chlorobenzene basic unit contents of 0%, 12.5%, 25%, 35% and 50%. The mechanism of adsorption of carbon disulfide by hydroxyl-modified activated carbon was studied. It was found that at low pressure, when the basic unit content of hydroxyl-modified hexabenzobenzene was 50%, its adsorption capacity for carbon disulfide molecules was the highest, and when the basic unit content of hydroxyl-modified hexabenzobenzene was 25%, its adsorption efficiency for carbon disulfide molecules was the highest. The present manuscript is suitable for publication in Molecules Journal, subject to the following minor revision points:

1)      Authors should better highlight a crucial details of the Monte-Carlo (MC) simulation employed in this work, such as a number of simulations, the main assumptions, limitations of MC approach, etc.

2)      Authors explain the basics of the molecular dynamics approach used in this work, e.g.  by adding the most important equation(s)).  

3)      Authors should mention in the introduction some diffusion modeling papers, such as:

A comparative study of two different finite difference methods for solving advection–diffusion reaction equation for modeling exponential traveling wave, Ricerche di Matematica, Vol. 71, 2022, pp. 245–252.

4)      English should be carefully checked throughout the manuscript.

Minor editing of English language required

Reviewer 3 Report

The paper deals with the establishment of the structural model of activated carbon for the adsorption of carbon disulfide using grand canonical Monte Carlo simulation and molecular dynamics simulation. Five activated carbon models were proposed and used for the adsorption of carbon disulfide and the study include the change in pore volume and porosity, interaction between adsorbate and adsorbent, adsorption position, kinetics. The results can be considered a helpful reference for developing activated carbon for the carbon disulfide adsorption. The scientific of paper is good and before publication this paper must be improved along the following guidelines:

* The authors need to cite other articles in the introduction about preparation and use of activated carbon: for example: Heliyon 8 (2022) e11940; Processes 8 (2020) 1651; Chemosphere 294 (2022) 133764; J. Environ. Chem. Eng. 9 (2021) 105905; J. Environ. Chem. Eng. 7 (2019) 102775; J. Energy Storage 54 (2022) 105290; Chem. Afr. 6 (2023) 683–698.

* Page 1 L42: "The authors E.F. Spink[5] et al…."  -->  "Spink and Muller [5]…"

* Page 2 L47: "According to T. Morooka [6] et al.,.."  -->  "According to Morooka et al. [6],.." 

* Page 2 L82: "For example, Barris Demir [20] and others thought…"  -->  "For example, Demir and Ahunbay [20] and others thought……"

* Page 2 L87-88: "Daniel Bahamon [21] et al. …"  -->  "Bahamon and Vega [21] …" 

* Authors must revise the references section and respect the instructions of the journal.

Number of errors throughout the manuscript.

Reviewer 4 Report

This study uses GCMC and MD simulations to construct models of activated carbon with varying levels of hydroxyl-modified hexachlorobenzene basic units. The authors investigated the adsorption mechanism of carbon disulfide by hydroxyl-modified activated carbon and find that the adsorption capacity and efficiency of carbon disulfide vary with the content of hydroxyl-modified hexachlorobenzene basic units. Changes in the porosity, accessible surface area, and pore diameter of the activated carbon model lead to differences in the diffusion coefficient of carbon disulfide molecules in different hydroxyl-modified activated carbons. 

The work is very interesting, well organized and at the heart of the journal's topics. I therefore recommend its publication once the following questions/comments are addressed:

1. How can we be sure that the lowest energy adsorption configuration obtained through the Sorption module accurately represents the actual adsorption configuration of carbon disulfide on activated carbon models?

2. Are there any limitations to the GCMC simulation method that may impact the accuracy of the results obtained in this study?

3. DFT calculations would provide additional information about the adsorption behavior of the activated carbon models, such as the electronic structure and bonding interactions between the carbon disulfide molecules and the activated carbon surface. This information could be used to further analyze and understand the adsorption behavior observed in the GCMC simulations. I recommend comparing the current results with other DFT reports from the literature and try to explain how this work differs: 10.1016/j.susc.2021.121860, 10.1007/s10008-017-3703-3, 10.1016/j.seppur.2020.118086.

Round 2

Reviewer 2 Report

No comment

Author Response

We didn't see a clear second round of comments, so we didn't reply.